# Human Glucose Transporters in Health and Selected Neurodegenerative Diseases

**DOI:** 10.3390/ijms26157392

**Published:** 2025-07-31

**Authors:** Leszek Szablewski

**Affiliations:** Chair and Department of General Biology and Parasitology, Medical University of Warsaw, Chałubińskiego Str. 5, 02-004 Warsaw, Poland; leszek.szablewski@wum.edu.pl; Tel.: +48-22-621-26-07

**Keywords:** GLUT proteins, SGLT proteins, SWEET proteins, neurodegenerative diseases, Alzheimer’s disease, Huntington’s disease, Parkinson’s disease, GLUT1-deficiency syndrome, stroke, traumatic brain injury, therapies

## Abstract

Glucose is the main source of energy and the source of carbon for the biosynthesis of several molecules, such as neurotransmitters, for most mammalian cells. Therefore, the transport of glucose into cells is very important. There are described three distinct families of glucose transporters: facilitative glucose transporters (GLUTs), sodium-dependent glucose cotransporters (SGLTs), and a uniporter, the SWEET protein. Impaired function and/or expression of these transporters due to, for example, mutations in their genes, may cause severe diseases. Associations with the impaired function of glucose transporters have been described in the case of neurodegenerative diseases (NDs) such as Alzheimer’s disease, Parkinson’s disease, Huntington’s disease, GLUT1-deficiency syndrome, stroke, and traumatic brain injury. Changes in the presence of glucose transporters may be a cause of NDs, and they may be the effect of NDs. On the other hand, in many cases of neurodegenerative diseases, changes in the expression of glucose transporters may be a targeted therapy in the treatment of patients with these diseases.

## 1. Introduction

In most mammalian cells, glucose plays an important role as a source of energy for cellular functions and as a source of carbon for the biosynthesis of several biomolecules. For the appropriate delivery of glucose to the parenchyma, it must overcome four barriers. These barriers are associated with the absorption of glucose by the small intestine into the blood; its hepatic storage and release to the circulation; traversing the vascular endothelium for delivering sugar to parenchymal cells, for example, in the case of the blood–brain barrier (BBB); and reabsorption of glucose by the kidney back into the blood. The presence of these four cellular barriers means that glucose must be delivered from one compartment to another, from extracellular space to intracellular space and from intracellular compartment to extracellular compartment [1]. However, although the transport of glucose between components is necessary, it is difficult. This is due to the properties of glucose and the cell membrane. Glucose is a polar, hydrophilic molecule, whereas the cell membrane is made up of a lipid bilayer that is lipophilic and hydrophobic. Therefore, the cell membrane is impermeable to several cations (K^+^, Na^+^, Ca^2+^), anions (Cl^−^, HCO_3_^−^), hydrophilic molecules, as well as glucose. Therefore, to undergo diffusion through the lipid bilayer, glucose needs specific carrier proteins.

The transport of glucose across the cell membrane is caused by so-called “glucose transporters”, which transport glucose and several other substrates. In mammals, including humans, glucose is transported by a specific saturable transport process involving members of three different classes of glucose transporters: facilitative glucose transporters (GLUT proteins, *SLC2* genes), sodium-dependent glucose symporters (secondary active transport, SGLT proteins, *SLC5* genes), and a new class of glucose uniporters (SWEET proteins, *SLC50* genes) [2], each with different kinetic properties. Most cells express a variety of glucose transporters. The expression of glucose transporters in different tissues depends on the specific metabolic requirements. Disturbances in the function and/or expression of these transport proteins cause not only neurodegenerative diseases [3,4,5,6,7], but also several other human diseases, such as cancers [8], renal and heart diseases [9,10], and several others.

## 2. Characteristics of Human Glucose Transporters

Glucose transporters belong to the major facilitator superfamily (MFS). In the MFS, the sugar porter family is the largest group. These porters are detected in bacteria, archaea, and eukaryotes [11,12]. MFS contains 74 families of membrane transporters, each of which transports specific substrates. To date, more than 10,000 members have been sequenced. Proteins belonging to the MFS are ubiquitously expressed. They are also highly conserved from bacteria to humans. Glucose transporters exhibit different substrate specificities, kinetic properties, and tissue expression profiles.

### 2.1. The Human GLUT Transporters

In 1948, LeFevre suggested that for the transport of glucose across the lipid bilayer, a specific component within the cellular plasma membrane is necessary [13]. The human GLUT1 protein was purified in 1977 from erythrocytes [14], and the first GLUT protein to be isolated was cloned from HepG2 cell lines in 1985. The genetic and protein sequence of GLUT1 was performed in 1985 [15].

#### Structure of GLUT Proteins

All GLUT proteins contain 12 hydrophobic membranes spanning α-helical transmembrane (TM) domains, which are connected by the hydrophilic loop between TM6 and TM7 of the protein [16,17,18]. These glucose transporters contain a short intracellular N-terminal segment and a single site for glycosylation on the exofacial end. This site may be located in the large loop between TM1 and TM2 (first extracellular loop) or between TM9 and TM10 (fifth extracellular loop) [19]. Sequences of the GLUT family are better conserved in the putative TM regions, while in the loops between the TM domains and the C- and N-termini, sequences are more divergent. Among members of the GLUT proteins, the sequences are 14–63% identical and 30–79% conserved [20]. In all members of GLUTs are several highly conserved sequences, such as PMY in TM4, QQLSGIN in TM7, GPGPIP/TW in TM10, PESPRY/FLL in loop 6, GRR in loop 8, and VPETKG in the terminal tail. There are also 18 conserved glycine residues. Performed observations revealed that glycine residues in helices 1, 2, 4, 5, 7, 8, and 10 play an important role in the structure of these helices [20].

GLUT proteins are encoded by the solute-linked carrier family 2, subfamily A gene family [19,21,22]. In humans, 14 members of the GLUT family have been identified that are categorized into three classes according to sequence similarity [16,17,19,20,23,24]. Class I GLUTs include GLUT1–GLUT4 and GLUT14. These glucose transporters are 48–63% identical in humans. Specific residues for Class I GLUTs are the following: Glutamine in α-helical structure 5 (QL motif corresponding with Q161 in GLUT1) plays a role in the identification of glucose [25]. The STS (serine–threonine–serine) motif in extracellular loop 7 is involved in the conformational change during the transport process [26]. The highly conserved motif GPXXXP in TM10 is important for the binding of the two inhibitors cytochalasin B and forskolin, which affect the transport of glucose [27]. A QLS motif in TM7 is present in GLUT proteins that transport glucose but not fructose. This motif is associated with the selectivity of glucose and fructose [28]. Class II GLUTs comprise GLUT5, GLUT7, GLUT9, and GLUT11. These transporters are 36–40% identical. The lack of tryptophan residue following the conserved GPXXXP motif in TM10, corresponding with tryptophan 388 in GLUT1, and the absence of the QLS motif, are characteristic sequences for Class II proteins [28]. Class III GLUTs comprise GLUT6, GLUT8, GLUT10, GLUT12, and GLUT13 (HMIT). These transporters are only 19–41% identical. Many of the motifs detected in the other classes are conserved, such as PESPR in TM6, GRR in loops 2 and 8, PETKGR in TM12, arginine and glutamate residues in loops 4 and 10, and tryptophan residue after the GPXXXP motif in TM10 [21]. GLUTs of Class III do not contain the QLS motif, which is unique to the Class I GLUTs [22] (Figure 1). For more details, see [7].

All GLUT proteins are facilitative transporters, except HMIT, encoded by the *SLC2A13* gene, which is an H^+^/*myo*-inositol symporter [29]. In dependence on GLUT proteins, these transporters may play different roles (Table 1).

##### Pseudogenes

There are described four pseudogenes of the *SLC2A* family [17,19]:*SLC2A3P1* (alias GLUT6 or GLUT3 pseudogene) located on chromosome 5q35. It is a retroposon of *SLC2A3*.*SLC2A3P2* (alias GLUT3 pseudogene 2) located on chromosome 1p31.3. It is a retroposon of *SLC2A3*.*SLC2A3P4* (alias GLUT3 pseudogene 4) located on chromosome 8q21.3. It is a retroposon of *SLC2A3*.*SLC2AXP1* is located on chromosome 2q11.2. It contains internal stop sequences.

### 2.2. The Human Sodium-Dependent Glucose Cotransporters

The role of sodium in glucose transport was proposed in 1958 by Riklis and Quastel [115]. In 1962, Crane presented a hypothesis on sodium/glucose cotransport [116]. Crane also introduced the term “secondary active transport” (indirect active transport), because the hydrolysis of ATP is indirectly associated with the transport of glucose via the electrochemical gradient. According to the suggestion of Crane, the sodium-dependent glucose cotransporter contains two binding sites: one for glucose and one for sodium [117].

#### Structure of Sodium-Dependent Glucose Cotransporters

The sodium-dependent glucose cotransporters, also known as sodium/glucose cotransporters, are energy-dependent and function as active transporters. They transport glucose against the glucose concentration gradient, and they are dependent on sodium cotransport. One molecule of glucose is transported with one or two sodium ions. Sodium ions are transported down a sodium concentration gradient [5,118]. The sodium/glucose cotransporters belong to the gene family (*SLC5A*), the SGLTs or sodium/substrate symporters family (SSSF), that contains more than 450 members [119,120,121]. *SLC5A* genes encode for proteins of 580–718 amino acid residues with a predicted mass of 60–80 kDa. There are differences between the structures of the genes. In eight genes, the coding sequences are in 14–15 exons (*SLC5A1*, *SLC5A2*, *SLC5A4*–*SLC5A6*, and *SLC5A9*–*SLC5A11*). The coding sequence for *SLC5A3* and for *SLC5A7* are present in exons 1 and 8, respectively. In the *SLC5A3* and *SLC5A9*–*SLC5A11* genes, there is alternative splicing. Performed investigations suggest that all of these symporters contain 14 TM α-helices (TMHs). The sodium/iodide symporter (NIS) and SMCT1 lack TMH^14^ [122]. Both the N- and C-termini are located on the extracellular side of the plasma membrane [2] (Figure 2). There are variable numbers for N-linked glycosylation. However, SGLTs are highly glycosylated membrane proteins; glycosylation is not required for their function. In humans, there are 12 genes that encode these symporters, which are expressed in different tissues. All of these genes code sodium/glucose cotransporters, except for SGLT3, encoded by the *SLC5A4* gene, which plays a role as a glucose sensor [123]. Six genes of this family encode cotransporters for solute, such as glucose, *myo*-inositol, and iodide; one is a Na^+^/Cl^−^ choline cotransporter and another is a glucose-activated ion channel [121]. In the dependence on sodium-dependent glucose cotransporter proteins, these transporters may play different roles (Table 2).

### 2.3. The Human SWEET Transporter

SWEET proteins are involved in the transport of mono- and disaccharides across the vacuolar and plasma membranes. These transporters have seven predicted TM domains with two internal triple-helix bundles (THBs) connected by an inversion linker helix (TM4), creating the translocation pathway (3 + 1 + 3 configuration). This class of transporters was first identified in *Arabidopsis* [131]. In *A. thaliana,* up to two dozen SWEETs have been identified, whereas animals usually have only one SWEET, except *Caenorhabditis elegans*, in which seven genes encoding SWEETs were found [42]. Human SWEET (RAG1AP1) is encoded by the *SLC50A1* gene located on chromosome 1q22 [156]. It contains 221 amino acids with a molecular weight of 25 kDa. SWEET1 does not stimulate glucose uptake, but instead mediates a weak efflux. Expression of human SWEET1 in HEK293T cells reveals its localization in the Golgi complex. Its higher expression was detected in the oviduct, epididymis, and intestine. In the mouse mammary gland, its expression was induced during lactation. It is suggested that in the basolateral membrane of enterocytes, SWEET may be associated with the exit of glucose from the cell into the blood in patients with Fanconi Bickel Syndrome [157].

## 3. Association of Human Glucose Transporters with Neurodegenerative Diseases

The brain consumes approximately 20% of the energy intake of the body. Nutrients and small molecules are exchanged by the brain and blood at the blood–brain barrier (BBB). The brain contains various transport systems for acquiring substances such as amino acids, lactate, ketone bodies, lipids, and cofactors for functions of neurons. Glucose plays an important role in the production of energy and as a source of carbon for biogenesis in neurons. Glucose is taken up from blood by glucose transporters, which are expressed in several cell types. As an alternative source of energy for neurons, lactate may be used [158,159], as well as ketones, by the capillaries of the brain and by astrocytes [160,161]. As mentioned earlier, 10 GLUTs and 10 sodium-dependent symporters are expressed in the CNS [3]. The expression of the SWEET1 protein in the human brain was not detected. The human brain expresses many transporters, but only a small number have been characterized [162]. However, in the human brain various isoforms of GLUTs and sodium-dependent cotransporters exist; a major role in fueling energy requirements is played by GLUT1 and GLUT3 [70].

### 3.1. Expression of Glucose Transporters in Alzheimer’s Disease

Alzheimer’s disease (AD) is a progressive neurodegenerative disease associated with severe and progressive impairment of cognitive function [163], memory loss, and language problems [164]. It may also include non-cognitive dysfunction (executive) that may be followed by behavioral disorders such as agitation, aggressiveness, and depression [163,164,165]. There are over 100 types of dementia, but the most well-known dementia is AD, which accounts for 50–75% of all cases of dementia [166]. Many observations have revealed that diabetes mellitus increases susceptibility to AD [167]; therefore, it has been suggested to use a new term for AD: type 3 diabetes mellitus (T3DM) [168,169,170,171]. Clinically, there are two types of AD [172]. About 90–95% of AD patients are aged ≥ 65 years and are diagnosed with “late onset” or “sporadic AD” (sAD) [173]. In 5–10% of patients, the disease is caused by a rare genetic mutation. These patients are diagnosed with “early onset” or “familial AD” (fAD). In fAD, symptoms of disease are detected in younger persons—those in their thirties, forties, or fifties [173,174,175,176]. Symptoms detected in fAD are caused by mutations in three genes: amyloid precursor protein (APP), presenilin-1 (PS-1), and presenilin-2 (PS-2). It is suggested also there are other mutations, but they are yet unknown [172]. In the case of sAD, the genetics are more complex [177]. Obtained results in performed observations showed that the epsilon 4 allele of the Apolipoprotein E 4 (ApoE-ε4) gene is a significant risk factor for the development of sAD. Two copies of the ApoE-ε4 gene increase the risk of AD 12-fold, and one copy increases the risk of AD 4-fold [178]. The mentioned gene is carried by 50–60% of individuals. It is suggested that other factors also influence the risk of developing AD [179].

The development of AD is associated with abnormal protein aggregation [180,181]. The proteinopathy symptoms observed in AD are associated with two types of misfolded protein in the brain: the neurofibrillary tangles (NFTs), containing hyperphosphorylated tau, and cellular plaques, with aggregated amyloid β (Aβ) peptides [182,183]. Impairment of synaptic functions begins the pathogenesis of AD. This pathology is caused by the accumulation of Aβ produced from the APP. APP is a transmembrane protein expressed in many tissues, including the CNS. The main sites of its concentration are the synapses of neurons, where it plays a role as a cell surface receptor. It is involved in the regulation of synapse formation, neural plasticity, antimicrobial activity, and export of iron [167,184]. APP is processed in different pathways. The first pathway, in which 90% of APP is processed, is the non-amyloidogenic (non-plaque-forming) pathway. The remaining 10% of APP is processed by the amyloidogenic pathway. In this process is involved γ-secretase. In fAD, its role is changed due to mutation in the APP, PS1, and PS2 genes, as well as the inheritance ApoE-ε4, resulting in the increased production of Aβ_42_, a neurotic isoform. Aβ peptides aggregate into oligomers and create fibrils in amyloid plaques, which block the signaling pathway and connection, causing cell death. Insulin plays an important role in the metabolism of APP. It regulates the balance between anabolism and catabolism of Aβ [167,184,185,186,187]. The tau protein is a neuronal microtubule-associated protein expressed in axons [188]. In the human brain are detected six isoforms of this protein caused by alternative splicing [189]. Tau is involved in the assembly and stability of microtubules, which are involved in processes such as morphogenesis, cell division, and intracellular trafficking. In neurons, tau is associated with synapses and nuclei [190]. The activity of tau is regulated by its phosphorylation. This protein contains more than 85 phosphorylated or phosphorylable sites, about 80 Ser/Thr and 5 Tyr phosphorylation sites [191,192]. The neuronal homeostasis is associated with kinases and phosphatases, which regulate the balance between tau phosphorylation and dephosphorylation. An imbalance between these enzymes may cause hyperphosphorylation of tau. If tau is phosphorylated at more than 30 sites, it may be involved in the development of Alzheimer’s diseases and NFT [188]. In the brains of patients with AD, tau protein is three times more phosphorylated than tau phosphorylation in brains of healthy subjects [193]. For more details, see [167].

In addition to Aβ and tau, AD is also a metabolic disease associated with reduced cerebral glucose metabolism, brain insulin resistance, and age-induced mitochondrial dysfunction [194,195]. It is suggested that glucose hypometabolism is a prominent feature of the brains of AD patients [196]. In sAD, the synthesis of ATP from the metabolism of glucose is decreased by 50%. Also observed is a tendency to decrease the synthesis of ATP throughout the progression of the disease [173,197]. Decreased uptake of glucose and glucose hypometabolism in specific brain areas are associated with extensive synaptic loss in advanced stages of AD [129,198,199]. It is suggested that hypometabolism in AD may be due to decreased glucose transport in the brain by glucose transporters and/or impaired brain metabolism [129]. Decreased glucose transport in the human AD-affected brain is observed in the most metabolically active brain regions such as the cortex, hippocampus, and cerebral microvessels [200,201,202]. Similar results were obtained also in animal studies [129]. The postmortem investigations of samples of the brains of AD patients revealed decreased levels of GLUT1 and GLUT3 proteins, mainly in the cerebral cortex, with significant loss of GLUT3 [203,204,205]. As mentioned earlier, GLUT1 and GLUT3 play a major role in glucose uptake by the brain; however, other GLUT proteins expressed in the brain also play a role in glucose transport in the brain. Decreased levels of GLUT1 and GLUT3 were observed in the hippocampal and cortical regions of AD patients [206] and in the frontal cortex [207]. Results obtained in another study also showed decreased levels of GLUT1 and GLUT3 in the brains of AD patients. Based on these results, the authors conclude that decreased levels of GLUT3 are related to the severity of AD pathology and the expression of AD symptoms [208]. Decreased levels of GLUT1 and GLUT3 were confirmed in animal models of AD [129,206,209]. Obtained results revealed also that decreased levels of GLUT1 and GLUT3 were correlated with decreased *O*-GlcNAcylation and abnormal hyperphosphorylation of tau [207,210]. Based on these results, it is suggested that deficiency of GLUT1 and GLUT3 in AD may impair brain glucose uptake/metabolism, causing neurodegeneration by the downregulation of *O*-GlcNAcylation and hyperphosphorylation of tau [207,210,211]. Decreased levels of the transcription factor hypoxia-inducible factor 1 (HIF-1) also were found [207]. HIF-1 is a major regulator of GLUT1 and GLUT3, and it is important in normal glucose metabolism [212]. Based on the obtained results, the authors suggest that a decrease in the mentioned glucose transporters may be due to downregulation of HIF-1 in the AD brain [207]. There are also other suggested mechanisms. For example, it is suggested that glucose hypometabolism caused by decreased levels of GLUT1 and GLUT3 contributes to abnormal tau hyperphosphorylation and/or neurofibrillary degeneration by hexosamine biosynthesis pathway downregulation [213]. Another mechanism associated with decreased levels of GLUT3, it is postulated that the promoter of human GLUT3 contains three potential cAMP response element (CRE)-like dements: CRE1, CRE2, and CRE3. Increased expression of GLUT3 may be associated with the overexpression of CRE-binding protein (CREB) or the activation of cAMP-dependent protein kinase. In the AD brain, decreased full-length CRB was detected, whereas truncation of CREB was increased. In the human brain, truncation is involved in the activation of calpain 1, which proteolyzes CREB, generating a truncated CREB. Truncated CREB has less activity to stimulate the expression of GLUT3. Performed investigations revealed in the human brain a positive correlation of GLUT3 with full-length CREB and a negative correlation with the activation of calpain 1. Overactivation of calpain 1 due to calcium and the overloaded proteolysis of CREB resulted in decreased expression of GLUT3. The effect is impaired glucose uptake and metabolism in the AD-affected brain [214]. Correlation between decreased levels of AD and Aβ deposition was confirmed in animal studies [215]. Also investigated was the expression of other glucose transporters in the AD brain. Obtained results revealed markedly increased (~1.5-fold) levels of GLUT2, whereas no change was detected in the level of GLUT4 [207]. Researchers observed that the increased level of GLUT2 in the brain is correlated to a similar extent with the activation of astrocytes. Therefore, it is suggested that the increased level of GLUT2 is correlated with the activation of astrocytes in AD [207]. In the frontal cortex of AD patients was detected a significantly high level of GLUT12 [216] and in the brain of mouse models of AD [217]. Unfortunately, there is insufficient information on this issue. Of note, in a *Drosophila* model of Aβ toxicity, increased glucose transport into neurons rescues Aβ toxicity [196].

### 3.2. Expression of Glucose Transporters in Parkinson’s Disease

Parkinson’s disease (PD), also known as paralysis agitans, is a common neurological disease in middle-aged and elderly people. It is the second most common neurodegenerative disease. In PD patients are observed resting tremor, bradykinesia, muscle rigidity, and postural instability [218]. The primary pathological change in PD is the degeneration and death of dopaminergic (DO) neurons, causing their loss in the substantia nigra (SN) pars compacta [219], and decreasing activity of the dopamine transmitter system in the nigrostriatal area. Closely associated with the viability of dopaminergic neurons are Lewy bodies, which are formed by the aggregation of α-synuclein (α-Syn). The cause of neuronal degeneration remains unclear; however, there are suggested to be multiple factors rather than a single cause, such as oxidative stress, mitochondrial dysfunction, neuroinflammation, and protein misfolding [220]. Most PD cases are sporadic (sPD), caused by a combination of genetic and environmental risk factors. Only 5% to 10% of PD may be due to a single pathogenic mutation (single gene) [221], and only about 15% are caused by heritable familial mutations [222]. There are suggestions that PD may be associated with changes in glucose metabolism in the frontal lobe and caudatum putamen [223]. Disorders of glucose metabolism resulting in energy metabolism failure may also be the cause of PD [224,225,226]. Unfortunately, the literature on the glucose transporters in PD is very scarce. Human induced pluripotent stem cells (hiPSCs) derived from fibroblasts of PD patients, and their differentiation product, neural precursor cells (hNPCs), were used in observations to expand the etiology of sPD. These cells were used as a human cellular model of sPD. Obtained results revealed that the levels of GLUT1 mRNA and GLUT3 mRNA were not altered in sPD hNPCs. Also, GLUT1 and GLUT3 proteins were not affected in these cells. The expression of the *SLC2A4* gene was significantly decreased, but the *SLC2A2* and *SLC2A4* genes were weakly expressed. Therefore, their role in glucose metabolism is questionable [227]. Experiments performed on a mouse model of PD revealed that density and localization of Glut1 were unaltered in these animals. The authors suggest that, although in PD there are metabolic disturbances, the level of Glut1 is not changed following dopaminergic neurodegeneration [228]. On the other hand, decreased levels of Glut1 in the striatum of a PD model in mice were reported [229,230]. Associations between glucose metabolism and Parkinson’s disease need further investigation.

### 3.3. Expression of Glucose Transporters in Huntington’s Disease

Huntington’s disease (HD) is a progressive, neurodegenerative disease caused by a CAG expansion in exon 1 of the huntingtin (HTT) gene, resulting in the formation of a prolonged polyglutamine (polyQ) tract in the N-terminal region, which leads to the production of a toxic mutant huntingtin protein (mHTT) [231]. PolyQ stretches of more than 36 residues produce a gain of function in mHTT, affecting the healthy function of cellular machinery, resulting in neurotoxicity and detrimental cell lethality [232]. In healthy individuals, the CAG region of HTT contains 11–35 trinucleotide repeats [6]. A fundamental pathogenesis of HD is abnormal accumulation of mHTT in the neurons. The misfolded and aggregated toxic protein propagates and spreads in a prion-like fashion [233]. The length of the CAG repeat inversely correlates with age of onset in patients with Huntington’s disease [234]. Mutations in the HTT gene cause the loss of critically important brain neurons such as medium spiny neurons in the striatum, a subcritical part of the forebrain. Disturbances of energy homeostasis in AD are associated with impaired glucose transporters, abnormal functions of glycolytic enzymes, etc. [235]. It was found that intracellular glucose levels in mHTT cells are between 0.8 and 1.5 mM. This is a 2.5-fold decrease in intracellular glucose levels in these cells, which may indicate a change in glucose metabolism. This suggestion was confirmed in a further study [236]. In HD patients, decreased uptake of striatal glucose was observed [237] in the caudate, putamen, and thalamic regions [238]. Reduced uptake of glucose was positively correlated with disease severity [238]. A loss in uptake of glucose restricted to the frontal and inferior parietal cortex was observed in newly diagnosed patients. A global loss of glucose uptake was detected in patients with symptoms of HD lasting longer than 5 years. In symptomatic patients and carriers of asymptomatic mHTT was observed a decreased uptake of glucose in the caudate nucleus and putamen that was correlated with the number of CAG repeat lengths [239]. Postmortem investigations of brain samples of patients with grade 3 HD revealed a decreased expression of GLUT1 and GLUT3 by three- and four-fold, respectively. On the other hand, similar observations in patients with earlier stages of HD (grade 1) revealed no changed expression of these GLUT proteins [240]. Results obtained from experiments performed on animal models of HD revealed a decreased uptake of glucose in the Huntington’s disease cortex caused by the loss of glucose transporters’ expression [241]. Results obtained in observations performed on animal models of HD may suggest that impaired glucose metabolism in patients with HD may be associated with impaired expression of the insulin gene [242]. In HD patients is observed not only decreased uptake of glucose, but also, increased uptake of lactate. This observation suggests that HD causes decreased levels of energy [243]. Mitochondrial metabolism disorders are observed in patients with HD and in animal models of this disease [244]. For example, there was observed in the postmortem brain samples of HD patients a loss of pyruvate dehydrogenase (PDH) and oxyglutarate dehydrogenase (OGDH), important enzymes in the Krebs cycle [245]. The N-terminal of mHTT interacts with mitochondrial membranes, resulting in the inhibition of respiratory complex II. Impaired mitochondrial electron transport causes the overproduction of reactive oxygen species (ROS) and decreased production of ATP [246]. Damage due to oxidative stress decreases the expression of GLUT3, causing stimulation of lactate uptake and inhibition of glucose uptake [247]. As mentioned above, in HD patients, the onset of HD is correlated with the copy number of the *SLC2A3* gene [234]. It was also observed that three copies of the *SLC2A3* gene may have a neuroprotective effect, and between the copy number of this gene and the level of GLUT3 expression is the linear correlation. These results, as well as those obtained in the *Drosophila* HD model, may suggest that an increased number of *SLC2A3* genes ameliorates HD pathologies. Because the progression of HD and the uptake of glucose by neurons mainly depends on GLUT3, it may have therapeutic ramifications. Observations performed on the *Drosophila* HD model revealed that an increased dosage of Glut1 also ameliorates HD phenotypes in these flies [234]. The beneficial role of increased levels of glucose transporters in HD pathology was described also by other authors [235,248]. However, disturbances in the function and expression of glucose transporters may contribute to the pathophysiology observed in HD; these transporters, in particular GLUT1 and GLUT3, may be involved with targeted therapy in the treatment of HD. It should also be investigated whether there is a correlation between HD and dysfunction of other glucose transporters. This problem needs further investigation.

### 3.4. Expression of Glucose Transporter in GLUT1 Deficiency Syndrome

GLUT1 deficiency syndrome (GLUT1-DS) is a rare genetic metabolic disease inherited in an autosomal recessive manner. It was first described as an early-onset childhood epileptic neuropathy [249]. Description of additional cases associated with mutations in the *SLC2A1* gene caused the phenotype spectrum to be expanded by epileptic encephalopathy with different types of seizures, movement disorders, and paroxysmal events of non-epileptic origin [250,251]. GLUT1-DS accounts for around 1% of idiopathic generalized epilepsies and approximately 10% of early-onset absence of epilepsies [252]. This neurological disease is associated with decreased expression and/or function of GLUT1 in the brain [249,251,253,254]. To date, there are more than 20 mutations described. In most cases, GLUT1-DS is due to heterozygous single-nucleotide variants (SNVs) in the *SLC2A1* gene. These mutations cause complete or severe impairment of functionality and/or expression of GLUT1 in the brain [255]. The SNVs cause amino acid exchanges, exon deletions, and frame shifts, influencing the regulation of transcription or translation [256,257,258,259]. Clinical neurological symptoms depend on the type of mutation. Missense mutations are predominant in mild and moderate clinical categories [260,261]. Insertion causes the most severe phenotype of GLUT1-DS [262]. Heterozygous forms of the *SLC2A1* gene, causing partial deficiency of GLUT1, are associated with intractable infantile epilepsy [262]. In these babies, the head is normal size, but they display mild chronic encephalopathy with infrequent seizures, spasticity, and ataxia. The lighter forms of GLUT1-DS are associated with hypoglycemia, clumsiness, speech delay, frequent perplexity, and loss of consciousness [263]. Results obtained in animal studies revealed that the recessive homozygous form is lethal [264,265]. This syndrome affects all ages. In patients with GLUT1-DS, other clinical manifestations are observed, such as myoclonic limb jerking with alternating staring and eye-rolling, unresponsiveness, head bobbing, generalized seizures, ataxia, spasticity dystonia, intermittent ataxia, periodic confusion, periodic weakness, and recurrent headaches. In these patients may be detected also specific, atypical manifestations of GLUT1-DS, such as intermittent ataxia, dystonia, migraine, etc. [250,251]. It is suggested that the pathologies observed in GLUT1-DS are due to an insufficient supply of glucose for neuronal activities [162]. For therapy of GLUT1-DS, ketonic or high-fat diets are prescribed to patients. Ketonic diets improve various, but not all, symptoms of GLUT1-DS. For example, this diet controls the seizures and other paroxysmal activities, but is less effective on cognitive functions [262]. In some patients, modified Atkins diets have been introduced [266]. Results obtained on animal models of GLUT1-DS are described in detail in [5].

### 3.5. Expression of Glucose Transporters in Stroke

Stroke is a devastating neurological disturbance that is the second cause of death worldwide. Performed observations revealed that more than 80% of stroke events are ischemic stroke, caused by restricted blood flow to a brain part. The most cases are due to occlusion as an effect of thrombosis, embolism, and/or arteriosclerosis [5]. Performed observations and experiments on animals revealed changes in the expression of glucose transporters and/or their mRNAs. In a rat model of median cerebral artery occlusion (MCAO), one hour after short-term MCAO, upregulation of GLUT1 mRNA was observed in the ipsilateral and contralateral brain cortex. Upregulation of GLUT1 in the investigated rats was detected in microvessels, astrocytes, and distinct populations of neurons. Upregulation of GLUT1 mRNA was detected in glial cells of the penumbra, but not in neurons, one day after MCAO [267]. It was found also that in rats with the median cerebral artery (MCA) occluded for 3 h, upregulation of GLUT1 protein and GLUT1 mRNA was observed at 12 h of reperfusion in an ipsilateral cortical area outside the core infract region [268]. Upregulation of GLUT1 protein and GLUT1 mRNA as an effect of ischemia was described also by other researchers [269,270]. Based on the obtained results, it is suggested that in the regulation of the expression of GLUT1/GLUT1 during stroke, heat shock protein (HSP) 70, hypoxia-inducible factor (HIF) 1, and insulin-like growth factor-1 (IGF-1) are involved [271,272]. During stroke, GLUT3 is also upregulated; however, its regulation differs in comparison with GLUT1. No changes in levels of GLUT3 mRNA were observed in the ipsilateral or contralateral forebrain throughout the first couple of hours after short-term MCAO in experimental rats [267]. The slightly increased expression of GLUT3 mRNA in neurons of the ipsilateral cortex was observed one day after MCAO. At this time, the expression of GLUT1 mRNA subsided [268]. Associations of different expressions of GLUT3 protein and GLUT3 mRNA with ischemia were described also by other authors [270]. The regulation of GLUT3 expression in response to energy depletion during ischemia was investigated on cultivated neurons and astrocytes. Obtained results revealed that during the onset of ischemia, GLUT3 is rapidly inserted into the plasma membrane, which may protect neuronal cells against death in response to ischemia [273]. It also was observed that during ischemia, in cultivated porcine brain cells, Sglt1-mediated uptake of glucose across the BBM was increased [5]. For more details, see [5].

### 3.6. Expression of Glucose Transporters in Traumatic Brain Injury

Traumatic brain injury (TBI) comprises “… focal and diffuse brain damage caused by different types of violation and brain concussion” [5]. It may cause mental and physical disabilities. Following brain injury, patients have been observed to have changes in cerebral D-glucose uptake and expression of cerebral glucose transporters. Decreased uptake of D-glucose was found in patients during the acute phase of TBI. This observation may suggest decreased metabolism (hypometabolism) [274]. In patients after severe brain trauma, cerebral glucose uptake was increased after 1 week [275]. In brain samples obtained from patients about 8 h after TBI upregulation of 55 kDa GLUT1 expression was observed, predominantly located in the endothelial cells of brain capillaries [276]. Experiments performed on animal models revealed that in the brains of rodents, no effects on the expression of 45 kDa GLUT1 were observed up to 2 days after traumatic injury. The increased expression was observed in the case of 55 kDa GLUT1 6 h and 2 days after trauma [277]. Based on these results and others, there is suggested to be posttranslational upregulation of GLUT1/GLUT1 after TBI [5]. It is suggested that in the regulation of GLUT1/GLUT1 expression, HIF-α is involved [5,278]. In the case of GLUT3, it was found in observations of animals that distinct upregulation of this glucose transporter was detected in the cerebral cortex and cerebellum 4 h after TBI [279]. Upregulation of GLUT3 was still detected two days after TBI [5]. Only very limited observations are available in the case of the expression and function of SGLT1/Sglt1 in the brain during TBI. Obtained results from animal studies suggest that upregulation of SGLT1/Sglt1 during TBI aggravates secondary tissue damage and clinical outcomes [5].

## 4. Therapeutic Strategies in Neurodegenerative Diseases

Epidemiological studies showed a potential link between diabetes mellitus and neurodegenerative diseases (NDs) such as AD, PD, HD, and amyotrophic lateral sclerosis (ALS) [280]. In these diseases are observed pathological features similar to those observed in DM, such as mitochondrial dysfunction, insulin resistance, impaired insulin signaling, oxidative stress, etc. There are also pathologies specific to particular NDs. Diabetic patients have an increased risk of developing NDs. However, in the case of association between DM and PD, there are different results. For example, case-control studies suggest a negative association between DM and PD [281,282]. On the other hand, meta-analyses of prospective cohort studies suggest an increased risk of PD development in diabetic patients [283,284]. Similarity in the pathological features detected in the mentioned diseases means that the therapeutic strategies for NDs are based on anti-diabetic drugs [285,286,287]. However, the exact mechanisms remain not fully known; these drugs may play a novel role in managing neurodegenerative diseases (Table 3). Most of these drugs show a beneficial effect on NDs. On the other hand, it must be added that some of these drugs only alleviate certain symptoms rather than providing a cure.

### 4.1. Incretins

Glucagon-like peptide-1 (GLP-1) is an endogenous multifunctional peptide secreted by intestinal L cells in response to food intake. It enhances insulin secretion by pancreatic β cells and regulates blood glucose, food intake, and body weight [324,325]. GLP-1 is also synthesized in the brain by hypothalamic neurons within the nucleus of solitary tract, intermediate reticular nucleus, pyriform cortex, and olfactory bulb [300]. GLP-1 receptors (GLP-1Rs) are mainly expressed in the intestine and the α and β cells of the pancreatic islets. GLP-1Rs are expressed also in several CNS regions, such as the hippocampus, cortex, hypothalamus, and other areas, which are associated with AD and PD [326,327]. GLP-1R triggers cyclic adenosine-monophosphate (cAMP)-dependent PI3K/AKT and PKA/MAPK pathway activation. These pathways modulate mitochondrial function, glucose homeostasis, and apoptosis, and they are associated with processes such as inflammation, satiety, memory, synaptic plasticity, neurogenesis, and stress response [328]. To date, most GLP-1R agonists, such as liraglutide, semaglutide, lixisenatide, albiglutide, and dulaglutide, are used in therapy in diabetic patients. Results obtained in investigations of GLP-1R performed through tests in vivo and in vitro in cell lines, animal models of NDs, and ND patients revealed neuroprotective properties [288,289,290,291,292,293,294,295,296,297]. Exendin-4 is a peptide agonist of GLP-1R that has been detected in the saliva of the Gila monster (*Heloderma suspectum*). The synthetic version of exendin-4 revealed a 53% amino acid sequence homology with human GLP-1 [329]. Treatment by exenatide halts the cognitive decline in Aβ-induced rat and mouse models of AD [297,298]. It improves degeneration of dopaminergic neurons and motor function in mouse and rat models of PD [289,299]. Exendin-4 decreases mitochondrial toxicity in an Aβ-induced AD mouse model [290], alleviates cognitive dysfunction, prevents amyloid-β accumulation, and protects mitochondrial function in 5xFAD mice [297]. Liraglutide ameliorates learning and memory impairment in various rat and mouse models of AD. It attenuates mutant huntingtin-induced neurotoxicity in human neuronal cells [292,293,294,295]. GLP-1R agonists may promote neuroprotection against ischemic stroke [296].

Glucose-dependent insulin-releasing polypeptide (GIP), also known as gastric inhibitory, is a gut hormone involved in insulin secretion in response to food intake. Its receptors (GIPRs) are expressed in the neurons of the olfactory bulb, hippocampus, cerebellum, cerebral cortex, amygdala, substantia nigra, thalamus, hypothalamus, and brainstem. These receptors play an important role in neurotransmission, neurogenesis, and neuroplasticity [330,331]. Several experiments revealed that GIPR agonists have neuroprotective properties in mouse models of AD and PD [300,301,302]. These agonists reduce the activation of chronic inflammatory response in the brain, oxidative stress, synapse loss, amyloid plaque burden, and DNA damage, and they reduce lipid peroxidation and α-synuclein levels, protecting dopaminergic neurons in the substantia nigra [303,304]. Novel GLP-1R and GIP receptor agonists are being developed as novel therapy. These dual-receptor agonists of GIP and GLP-1 showed promising effect in animal models of AD and PD [300,301]. They activate both GIP and GLP-1 receptors. Obtained results showed that dual-receptor agonists rescue or prevent spatial learning and memory dysfunction [305,306,307], reduce amyloid plaques and phosphorylated tau [305], upregulate p-PI3K and p-Akt growth factor kinases, and prevent overactivation of p-GSK3β [307]. Other investigations observed in animal models of PD that dual agonists can improve motor function, protect dopaminergic neurons, and rescue glial cell line-derived neurotrophic factor levels [332]. Recently, triple-receptor agonists have been developed, activating GLP-1, GIP, and glucagon (Gcg) receptors (GLP-1/GIP/Gcg receptors). In mouse models of AD, the observed effects of triple-receptor agonists were anti-apoptotic effects, reduced Aβ deposition and hyperphosphorylated hippocampal tau, protection from loss of synapses, and reduced inflammatory and oxidative stress [308,309,310]. In PD mice, the triple-receptor agonist protected against dopaminergic neuronal death, decreased α-synuclein accumulation, and prevented nigrostriatal neurodegeneration [311,312].

### 4.2. Dipeptidyl Peptidase-4 Inhibitors

Dipeptidyl peptidase-4 (DPP-4) inhibitors influence the enzymes causing the inactivation of incretin, such as GLP-1. The inhibition of DPP-4 increases levels of GLP-1 in the blood. In animal models of AD, linagliptin improves cognitive function and decreases tau phosphorylation and Aβ aggregation [313,314]. Linagliptin has a neuroprotective effect on human nerve cells, because it protects neurons from the effect of Aβ on mitochondrial damage, oxidative stress, and impairment of the insulin signaling pathway. Unfortunately, the positive effects of DPP-4 inhibitors in animal models of NDs were obtained with higher doses than the ones used in human treatments. Based on these results, it is difficult to extrapolate the results to humans. On the other hand, regarding data in humans, in individuals who receive DPP-4 inhibitors, the incidence of PD is significantly decreased [315,316].

### 4.3. Thiazolidinediones

Thiazolidinediones such as pioglitazone, rosiglitazone, and troglitazone increase insulin sensitivity by binding to the peroxisome proliferator-activated receptor gamma (PPARγ). Experimental and clinical studies revealed the neuroprotective role of this class of drugs. These drugs exert neuroprotective effects in animal models of AD and PD, significantly improving behavior and motor responses [317,318,319]. Rosiglitazone improves spatial memory, tested in the Morris water maze, and increases removal of Aβ deposits in AD mice [333]. There are conflicting results from clinical trials regarding the effect of rosiglitazone in the treatment of dementia. Some results indicate that rosiglitazone improves cognitive performance in AD patients [320], but other results do not confirm this effect of the drug [321].

### 4.4. Biguanides

Metformin, a drug that belongs in the drug class biguanides, is the main first-line medication for the treatment of type 2 diabetes. In animal models of PD, it decreases α-synuclein expression. In AD, metformin has a beneficial effect on cognitive function [322] by promoting neurogenesis and preventing amyloid deposition and tau phosphorylation [334,335]. The role of metformin in neurodegenerative diseases is not clear. It was found that metformin may stimulate the production of Aβ [336,337,338]. In diabetic patients treated with metformin is found an increased risk of cognitive impairment [337]. A meta-analysis that studied the effect of metformin on neurodegeneration revealed that this drug has no effect on NDs [323]. But in the case of PD, a significantly higher incidence of PD was observed in patients treated with metformin in comparison with non-metformin patients [323]. The associations between metformin and PD need further investigation [339]. Metformin also regulates the activity of astrocytes in PD [340].

## 5. Conclusions and Perspectives

Glucose transporters play an important role in human health. They are involved in transport not only of glucose, but also several other substances into and out of cells. Therefore, their expression is exactly controlled. Disturbances in their expression and/or function are associated with different diseases, including cancers and metabolic diseases. Changes in the expression of glucose transporters are observed also in neurodegenerative diseases such as Alzheimer’s disease, Huntington’s disease, stroke, etc. Therapy of these diseases is based on anti-diabetic drugs. The beneficial effects of these drugs, as well as potentially negative effects, have been described. Unfortunately, there is a lack of investigation of the associations between glucose transporters and used therapeutic strategies in NDs. Maybe this association will be helpful in the treatment of ND patients. Significant clinical data are required. Because our understanding of these associations remains limited, further research is necessary. Research is underway to investigate the mechanism underlying the neuroprotective effects of anti-diabetic drugs to create effective strategies for the prevention and treatment of NDs. The deficiency of insulin and its impaired signaling contributes to dysregulation in brain function. Activation of PI3K signaling can boost Insulin Degrading Enzyme, showing promise against NDs. To validate the use of anti-diabetic drugs such as those mentioned above for the treatment of NDs, significant clinical data are required. Unfortunately, our understanding of these therapies remains limited. Therapies should include measures that increase the insulin supply and enhance INSR responsiveness and insulin-targeted gene expression. Animal models of NDs, especially mice and rats, used in preclinical studies have provided substantial insights into NDs’ pathogenesis. There are differences between the results obtained in these animal studies and clinical presentations. Therefore, the use of non-human primate models in these investigations may enhance the translational relevance of preclinical findings and validate novel therapeutic strategies. It is also important to enhance collaborations between experimental researchers and clinical practitioners to ensure the effective translation of obtained results into specialized treatment and prevention methods.

## Figures and Tables

**Figure 1 ijms-26-07392-f001:**
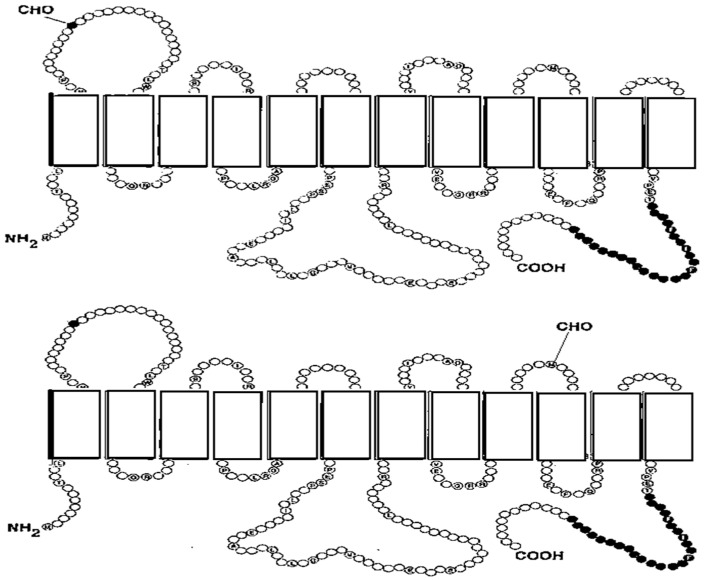
The N-linked glycosylation sites in Class I and Class II of glucose transporters (GLUTs) (**upper panel**) and in Class III (**lower panel**) [19].

**Figure 2 ijms-26-07392-f002:**
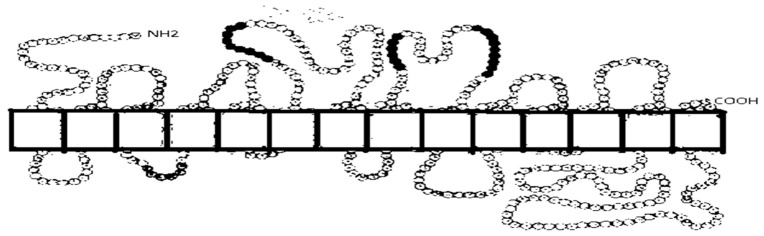
Structure of the human sodium-dependent glucose cotransporter (SGLT) protein. Presented site of N-glycosylation is for SGLT1 [120].

**Table 1 ijms-26-07392-t001:** General characteristics of human glucose transporters (GLUTs).

Glucose Transporter Gene Localization	Organ/Tissue of Transporter Expression	Transported Substrates	References
GLUT1 *SLC2A1* 1p35-p31.3	The large-molecular-weight GLUT1 (55 kDa) is detected in the cerebral cortex microvessels; the low-molecular-weight isoform (45 kDa) is present in the microvessel-depleted membrane of the brain (neuronal/glial membranes) and synaptosomes; an intermediate isoform of this transport protein is detected in the choroid plexus. It is expressed in fetal cells, from oocyte to the blastocyst. This transporter transports glucose through the epithelial or epithelial-like barrier cells of the brain, eye, peripheral nerve, and placenta. These barriers express high levels of GLUT1. Its expression is detected also in the blood cells, such as erythrocytes, granulocytes, and lymphocytes, as well as in adipocytes, muscle cells, kidney cells, and colon. Expression of GLUT1 was also detected in mitochondria of the human kidney cells 293T.	Glucose, galactose, mannose, glucosamine, the oxidized form of vitamin C.	[19,30,31,32,33,34,35,36,37,38,39,40,41,42,43,44,45,46]
GLUT2 *SLC2A2* 3q26.1-26.2.	It is expressed in the intestinal absorptive epithelial cells, hepatocytes, pancreatic β cells, and in the kidney cells. In the epithelial cells, GLUT2 is expressed in the basolateral membrane. In the brain, it is expressed in the brainstem, thalamus, cortex, hypothalamus, and hippocampus. Low levels of GLUT2 mRNA are detected in the nuclei of the brain, such as the nucleus tractus solitatius, the motor nucleus of the vagus, the paraventricular hypothalamic nucleus, the lateral hypothalamic area, the arcuate nucleus, and the olfactory bulbs, as well as in the neurons of limbic areas and related nuclei.	Glucose, galactose, mannose, fructose, glucosamine.	[17,20,30,42,47,48,49,50,51,52,53,54,55,56,57,58,59,60,61]
GLUT3 *SLC2A3* 12p13.3	Its high expression is detected in the brain, especially in the neurons, while not in the glia. In the brain, its expression is detected in the hippocampus, gyrus, and temporal neocortex. It is also found in tissues and cells with high demand for glucose, such as spermatozoa, placenta, preimplantation embryos, fibroblasts, platelets, white blood cells, and retinal endothelial cells.	Glucose, galactose.	[7,16,17,19,30,32,33,57,62,63,64,65,66,67,68]
GLUT4 *SLC2A4* 17p13	Its preferential sites of expression are the skeletal muscle, cardiomyocytes, and adipose tissue. It is expressed also in the brain in the hypothalamus, cerebellum, cortex, and hippocampus.	Glucose, dehydroascorbic acid, glucosamine.	[20,57,69,70,71,72,73,74,75,76]
GLUT5 *SLC2A5* 1p36.2	At high levels, it is expressed in the apical membrane of enterocytes. GLUT5 mRNA levels increase with age, and its highest levels are found in the small intestine of adults. Its expression is found also in the small intestine (proximal region). GLUT5 mRNA is detected in the plasma membrane of mature spermatozoa, and GLUT5 mRNA and/or GLUT5 protein have been detected in the kidneys, adipose tissue, skeletal muscle, and in the brain. Its expression is also detected in the microglial cells of the brain, but only in microglia among cells of the mononuclear phagocyte system.	Fructose.	[16,17,19,20,30,33,50,56,57,77,78,79,80,81,82,83,84]
GLUT6 Formerly designated as GLUT9 *SLC2A6* 9q34	It is predominantly expressed in the brain, peripheral leukocytes, spleen, and germinal cells of the testis.	Glucose.	[16,17,20,85,86]
GLUT7 *SLC2A7* 1p36.22	Its primary expression is detected in the apical membrane of the small intestine (distal region) and colon. Its mRNA is detected in the testis and prostate.	Glucose, fructose.	[17,20,87,88,89]
GLUT8 Formerly designated as GLUTX1 *SLC2A8* 9q33.3	It is detected in endosomes, lysosomes, and membranes of the endoplasmic reticulum. GLUT8 mRNA is detected in the testis, cerebellum, adrenal gland, liver, spleen, and brown adipose tissue. In the brain, expression of GLUT8 is reported in the hippocampus, dente gyrus, amygdala, primary olfactory cortex, hypothalamic nuclei, and tractus solitaries in the blood–retinal barrier. It is also reported in the endometrium and endometrial adenocarcinoma and in blastocysts. It transports sugars or sugar derivatives through the intracellular membranes.	Glucose.	[16,19,20,57,90,91,92,93,94,95]
GLUT9 Formerly designated as GLUTX *SLC2A9* 4p15.3-p16	Due to the use of alternative promoters, the gene encodes two isoforms of GLUT9: the major isoform, i.e., GLUT9 or GLUT9a, or an alternative splice variant of GLUT9 mRNA that codes the GLUT9ΔN or GLUT9b isoform. GLUT9 is expressed in the liver, kidneys, intestine, leukocytes, and chondrocytes, whereas GLUT9b is detected only in the liver and kidneys. GLUT9a is expressed in the basolateral membrane of the epithelial cells, while GLUT9b is localized to the apical pole.	Urate, glucose, fructose.	[19,20,96,97,98,99,100,101,102]
GLUT10 *SLC2A10* 20q13.1	It is detected in insulin-sensitive tissues such as the skeletal muscle and heart. GLUT10 mRNA is detected in the liver, pancreas, placenta, and the kidneys, as well as in human adipose tissue (omental and subcutaneous), in human preadipocyte cell strains (SGBS), and in 3T3-L1 adipocytes.	Deoxy-D-glucose, D-galactose.	[17,20,24,103,104,105]
GLUT11 Formerly designated as GLUT10 *SLC2A11* 22q11.2	In humans, three isoforms of GLUT11 are expressed. These isoforms are due to the separate exons (exons 1A, 1B, and 1C) of the *SLC2A11* gene, which generate three variants of GLUT11 (GLUT11-A, GLUT11-B, and GLUT11-C). GLUT11-A is detected in the heart, skeletal muscle, and in the kidneys; GLUT11-B is expressed in the kidneys, adipose tissue, and the placenta; and GLUT11-C is expressed in the adipose tissue, heart, skeletal muscle, and pancreas. This transporter has no rodent ortholog.	Fructose, glucose.	[19,20,106,107]
GLUT12 Formerly designated as GLUT8 *SLC2A12* 6q23.2	It is detected in the heart, skeletal muscle, adipose tissue, prostate glands, kidneys, small intestine, and chondrocytes. In the human placenta, different expressions of GLUT12 have been detected. In the first trimester, this glucose transporter is detected predominantly in the syncytiotrophoblast, in the lesser extent in the villous cytotrophoblast, and in extra-villous trophoblast cells. At term, GLUT12 is detected predominantly in the villous vascular smooth muscle and stromal cells, whereas in the syncytiotrophoblast, it is not detected.	Glucose, galactose, fructose.	[16,20,21,24,108,109,110]
GLUT13 (HMIT) *SLC2A13* 22q12	GLUT13 (HMIT) is an H^+^/*myo*-inositol cotransporter. Its high expression is detected predominantly in the brain in the hippocampus, hypothalamus, cerebellum, and brainstem regions. In the brain, this transporter is expressed in neurons and in the glial cells. Low levels of GLUT13 are detected in the white and brown adipose tissue and in the kidneys. It uses a proton gradient that is a source of energy for movement of substrate.	*Myo*-inositol, inositol-3-phosphate.	[17,19,20,111,112,113]
GLUT14 *SLC2A14* 12p13.3	It is identified and has been cloned as a duplication of GLUT3. There are two alternatively spliced forms: the short form of GLUT14 (GLUT14-S) and the long form of GLUT14 (GLUT14-L). Both isoforms are detected in the human testis. The ortholog of GLUT14 is not detected in mice.	Glucose, fructose.	[114]

**Table 2 ijms-26-07392-t002:** General characteristics of the human sodium-dependent glucose cotransporters.

Glucose Cotransporter Gene Localization	Organ/Tissue of Cotransporter Expression	Transported Substrates	References
SGLT1 *SLC5A1* 22q12.3	It is expressed in the BBM of the mature enterocytes in the small intestine, trachea, prostate gland, heart, kidneys, mammary glands, liver, and in the endothelial cells of the luminal membrane of intracerebral capillaries. Its expression is detected in the cortical, pyramidal, and Purkinje neuronal cells of the brain; in neurons (hippocampal) and the cortical granule (pyramidal); and the BBB. It is suggested that in the brain, SGLT1 plays a role as a glucose receptor. Its mRNA is also detected in the uterus (cervix).	Water molecules, urea, fluid across the intestine; D-glucose; D-galactose.	[2,20,77,119,120,121,122,124,125,126,127,128,129,130,131,132,133,134,135,136]
SGLT2 *SLC5A2* 16p11.2	Its predominant expression is detected on the apical membrane of renal convoluted proximal tubules (S1 and S2 segments). SGLT mRNA is detected also in the mammary glands, liver, lungs, intestine, skeletal muscle, and spleen. It is suggested also to have a role in the heart and brain as the receptor of glucose.	Glucose.	[2,20,119,120,137,138,139,140,141]
SGLT3 Alias SAAT1 *SLC5A4* 22q12.3	Its expression is detected in the kidneys, uterus, testis, intestinal autonomic nervous system, skeletal muscle, brain, the cholinergic neurons in the enteric nervous system, pancreas, lungs, and liver.	Sodium absorption in kidneys.	[20,119,121,131,142,143]
SGLT4 *SLC5A9* 1p32	Its expression is detected in the small intestine, kidneys, brain, liver, heart, uterus, and lungs. SGLT4 mRNA is detected in the testis, pancreas, and skeletal muscle.	D-mannose, D-fructose, D-glucose.	[2,104,119,121,131,144]
SGLT5 *SLC5A10* 17p11.2	SGLT5 is expressed in the human kidney cortex, and its mRNA is detected in the left atrium of the heart, ovary, skin (foreskin), testis, and vas deferens. Its precise location on the chromosome remains unknown.	Mannose, fructose, α-methyl-D-glucopyanoside.	[121,131,145,146]
SGLT6 SMIT2 *SLC5A11* 16p12.1	SGLT6 protein and its mRNA are detected in the brain, heart, kidneys, skeletal muscle, spleen, liver, placenta, lungs, and leukocytes. The *SLC5A11* gene reveals interaction with immune-related genes, and therefore, it is suggested that it may play a role as an autoimmune modifier gene.	*Myo*-inositol, D-chiro-inositol, D-glucose, D-xylose.	[121,131,146,147,148,149]
SMIT1 *SLC5A3* 21q22.11	There are three transcripts, SMIT1-1, SMIT1-2, and SMIT1-3, caused by splicing within and distal to exon 2. Its expression is detected in the kidneys, choroid plexus, brain, placenta, pancreas, heart, skeletal muscle, and lungs. SMIT1 mRNA is found in the blood vessels, choroid plexus, and uterus (cervix).	*Myo*-inositol, L-fucose, L-xylose.	[2,131,146,150]
NIS *SLC5A5* 19p13.11	It is a sodium/iodide cotransporter mainly expressed in the thyroid as well as in the lactating breast, colon, stomach, and ovaries.	I^−^, ClO_4_^−^, SCN^−^, NO_3_^−^, Br^−^.	[2,151,152]
SMVT *SLC5A6* 2q12	It is detected in the brain, heart, kidneys, lungs, and placenta. SMVT is a multivitamin/sodium cotransporter.	Vitamins such as pantothenic acid, biotin, α-lipoic acid.	[2]
CHT1 *SLC5A7* 2q13	It is expressed in the central nervous system, spinal cord, and medulla.	Choline.	[2,153]
SMCT1 *SLC5A8* 12q23.1	Its expression is detected in the small intestine, kidneys, brain, retina, and muscle.	Lactate, pyruvate, nicotinate.	[2,154]
SMCT2 *SLC5A12* 11p14.2	Its expression is detected in the small intestine, kidneys, brain, retina, and muscle.	Lactate, pyruvate, nicotinate.	[2,154,155]

**Table 3 ijms-26-07392-t003:** Comparison and characteristics of medications used in pharmacologic treatment of NDs.

Anti-Diabetic Drugs	Mechanism of Action and Effects [References]
Incretins	Glucagon-like peptide-1 (GLP-1) receptor agonists: Liraglutide reveals neuroprotective properties in ND patients and ND animal models [288,289,290,291]. It ameliorates learning and memory impairment in animal models of AD. Liraglutide attenuates mutant huntingtin-induced neurotoxicity in human neuronal cells [292,293,294,295]. These agonists promote neuroprotection against ischemic stroke [296]. Exenatide halts cognitive decline in animal models of AD [297,298] and improves degeneration of dopaminergic neurons and motor function in PD animal models [289,299]. Exendin-4 decreases mitochondrial toxicity, alleviates cognitive dysfunction, prevents Aβ accumulation, and protects mitochondrial functions [297,298]. Glucose-dependent insulinotropic polypeptide (GIP) receptor agonists: These agonists have neuroprotective properties in animal models of AD and PD [300,301,302] and reduce the activation of chronic inflammatory response in the brain, oxidative stress, synapse loss, amyloid plaque burden, and DNA damage. They reduce also lipid peroxidation and α-synuclein levels, protecting dopaminergic neurons in the substantia nigra [303,304]. Dual-receptor agonists of GLP-1 and GIP, which activate both GLP-1 and GIP receptors, show promising effects in animal models of AD and PD [300,301]. These novel agonists rescue or prevent spatial learning and memory dysfunction [305,306,307], reduce amyloid plaques and phosphorylated tau protein [305], and prevent overactivation of p-GSK3β. In animal models of PD, these agonists improve motor function and protect dopaminergic neurons. Triple-receptor agonists activate GLP-1, GIP, and glucagon (Gcg) receptors. Treatment of a mouse model of AD with these agonists revealed anti-apoptotic effects and reduced Aβ deposition and hyperphosphorylated tau protein. These agonists protect against loss of synapses and reduce inflammatory symptoms and oxidative stress [308,309,310]. In PD mice, the triple-receptor agonist protects against dopaminergic neuronal death, decreases α-synuclein levels, and prevents nigrostriatal neurodegeneration [311,312].
Dipeptidyl Peptidase-4 Inhibitors	Linagliptin improves cognitive function and decreases tau phosphorylation and Aβ aggregation in animal models of AD [313,314]. It has neuroprotective effects on human nerve cells caused by the protection of neurons from the effect of Aβ on mitochondrial damage, oxidative stress, and impaired insulin signaling. In humans who receive DPP-4 inhibitors, the incidence of PD is significantly decreased; however, high doses of inhibitors must be used [315,316].
Thiazolidinediones	This class of drugs reveals neuroprotective effects in animal models of AD and PD, significantly improving behavior and motor responses [317,318,319]. Rosiglitazone improves spatial memory tested in the Morris water maze. Treatment of AD patients with rosiglitazone revealed different results in the case of dementia. Some results indicate improved cognitive performance in these patients [320], while other results do not confirm this effect [321].
Biguanides	Metformin decreases α-synuclein expression in animal models of PD, and in AD animal models, it has a beneficial effect on cognitive function [322]. This effect is due to the promotion of neurogenesis and prevention of amyloid deposition and tau phosphorylation [320,321]. Performed observations revealed that metformin may have a negative effect in patients. This drug may stimulate the production of Aβ, and in diabetic patients treated with metformin, a significantly higher incidence of PD was observed in comparison with non-metformin diabetic patients [323]. A meta-analysis that studied the effects of metformin on neurodegeneration revealed that this drug has no effect on NDs [323].

## Data Availability

Not applicable.

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
