# Peer review of "Human Glucose Transporters in Health and Selected Neurodegenerative Diseases"

_ijms, 2025, doi:10.3390/ijms26157392_

Round 1

Reviewer 1 Report

Comments and Suggestions for Authors

The submitted manuscript could be of interest if focused on the role of human glucose transporters for neurodegenerative diseases and disorders. The authors give a huge place for the detailed description of the characteristics of human glucose transporters that have been extensively described in previous publications, including from the author. The narrative form of the main text  is also duplicated within the tables (column : organ/tissue of transporter expression). The reviewer considers that  section 2 is not essential for the understanding of the interesting part of the manuscript - sections 3 and 4 - and should be removed. If the authors decide to maintain section 2, it should be simplified and make reference to previous work. For sections 3 and 4 it will be necessary to add tables and figures to facilitate the reading since the current presentation is too difficult to follow. Probably use tables similar to table 2 and table 3 but with a simplification of the narrative part (use a few words) and the addition of relevant references. (table 1 with the names and codes of amino acids is not necessary).

The conclusion and perspectives section should be developed and include facts, not only speculation.

Author Response

Dear Reviewer

Reviewer 2 Report

Comments and Suggestions for Authors

This review article focuses on the role of glucose transporters in neurodegenerative diseases (NDs). It provides a comprehensive overview of the three major families of glucose transporters—GLUTs, SGLTs, and SWEET proteins—and discusses their physiological functions, particularly in the brain. The review highlights the altered expression or dysfunction of these transporters in various neurodegenerative conditions, such as Alzheimer’s disease, Parkinson’s disease, and Huntington’s disease, suggesting a potential link between glucose transport and disease pathogenesis. Moreover, the paper touches on the therapeutic implications of targeting glucose transporters, especially with anti-diabetic drugs. The article is informative and addresses a relevant topic, but there are several structural and content-related issues that need to be addressed to improve its clarity, coherence, and scientific rigor.

There are some issues:

  1. The abstract mentions neurodevelopmental disorders such as autism spectrum disorder; however, these are not discussed in the main text. It is recommended to revise the abstract to better reflect the content of the manuscript.
  2. The final sentence of the Introduction includes a list of diseases that are not further elaborated on in the manuscript. This sentence seems unnecessary and only increases the number of references without adding meaningful content.
  3. Table 1 appears to be redundant and does not add value to the manuscript. It is suggested that this table was removed.
  4. Since the manuscript describes the structure of GLUT proteins, it would be more informative to include a schematic figure illustrating their structure, rather than relying solely on text. The same suggestion applies to the descriptions of SGLT and SWEET proteins.
  5. Several sentences are broken prematurely due to unnecessary line breaks (e.g., Lines 89, 528). Please correct these formatting issues.
  6. Table 2 is overly detailed and lacks conciseness. Consider adding more columns to organize the data more effectively. The same applies to Table 3.
  7. The title of Section 3 is appropriate, but the use of both "diseases" and "disorders" is redundant. Choose one term for consistency and clarity.
  8. Some sentences have grammatical or logical issues and require revision, particularly those found in Lines 551–553, 639–641, 669–672, 690–691, 698–699, 710–711, 782–783, and 829–831.
  9. In Section 3, the discussion on Alzheimer’s disease (AD) and Parkinson’s disease (PD) is overly extensive, which weakens the focus on the role of glucose metabolism in disease mechanisms. Additionally, the manuscript does not adequately explore the causal relationship between glucose transport and disease progression. However, the subsection on "Expression of Glucose Transporters in GLUT1 Deficiency Syndrome" is well-written and informative.
  10. Regarding Section 4, it is recommended to include a figure or table summarizing the therapeutic agents mentioned. It should also be clearly stated that most of these drugs only alleviate certain symptoms rather than providing a cure.

Author Response

Dear Reviewer

Round 2

Reviewer 1 Report

Comments and Suggestions for Authors

The reviewer recognise the effort done to revise the manuscript and so improve the final layout, and also would like to thank the authors for that. Since all the concerns were addressed, the reviewer considers that the manuscript is now acceptable for publication in ijms and recommends to accept it in the present form.

Reviewer 2 Report

Comments and Suggestions for Authors

There are no outstanding issues.